# Deterministic and Random Generalized Complex Numbers Related to a Class of Positively Homogeneous Functionals

**Wolf-Dieter Richter**

Institute of Mathematics, University of Rostock, 18051 Rostock, Germany; wolf-dieter.richter@uni-rostock.de;
Tel.: +49-381-498-6551

**Abstract:** Based upon a new general vector-valued vector product, generalized complex numbers with respect to certain positively homogeneous functionals including norms and antinorms are introduced and a vector-valued Euler type formula for them is derived using a vector valued exponential function. Furthermore, generalized Cauchy–Riemann differential equations for generalized complex differentiable functions are derived. For random versions of the considered new type of generalized complex numbers, moments are introduced and uniform distributions on discs with respect to functionals of the considered type are analyzed. Moreover, generalized uniform distributions on corresponding circles are studied and a connection with generalized circle numbers, which are natural relatives of $\pi$, is established. Finally, random generalized complex numbers are considered which are star-shaped distributed.

**Keywords:** positively homogeneous functional; star body; vector-valued vector product; generalized complex multiplication; generalized complex division; vector-valued exponential function; Euler-type formula; complex algebraic structure; generalized complex plane; generalized complex differentiation; generalized Cauchy–Riemann differential equations; random generalized complex number; moments; uniform probability distribution; generalized uniform distribution on a generalized circle; uniform basis; generalized circle number; star-shaped distribution; generalized polar representation; stochastic representation

**MSC:** 11H99; 11L03; 60E05; 62E99

## 1. Introduction

Well-known generalizations of complex numbers include quaternions [1,2], octonions [3,4], bicomplex and multicomplex numbers [5,6] and Clifford algebras [7,8], among others. All of these works have a methodological common ground, which differs significantly from that used in the present work. Namely, products of elements of such structure are explained by the fact that certain expressions in brackets are formally treated as when multiplying expressions in brackets of real numbers, with additional assumptions being made for the multiplication of so-called basic elements. In contrast to this, suitable vector-valued vector products are introduced and used in the present work, which are geometrically well motivated as rotations and stretches.

Another essential difference between the present work and the group of works mentioned above is that the latter usually do not provide any information about which concrete mathematical objects fulfill the formulated wishes with regard to multiplication and whether the fulfillment of these wishes is unequivocal or ambiguous, while in concrete objects they are always specified in the present work.

To be more concrete: the present work generalizes [9] in that it considers generalized complex numbers with respect to certain general positively homogeneous functionals which contain the $l_p$-functionals considered in [9] as just a particular case. In particular, this new general approach allows the consideration of complex algebraic structures with

respect to any norm or antinorm and at the same time leads to the simplification of proofs. Furthermore, we open a new research area with the definition of the generalized complex differentiability of functions and the derivation of a first result for this.

Complex random variables are natural generalizations of ordinary, i.e., real random variables and are studied in an extensive literature such as, for example, in [10–13]. The close connection between analytical and geometric aspects when considering complex numbers is expressed in [14–17] among others, and was further developed in [9,18]. Probability densities for complex random variables are considered in [19], for the particular Gaussian case, see [20]. In [21,22] circularity is emphasized, while [23,24] consider complex elliptical distributions. The authors of [25–27] study statistical questions for complex random variables and [28] is devoted to statistics in the closely related field of shape analysis.

In [29], the authors refer to [30], where C.F. Gauss described the status of formation of complex numbers in the following words: "It could be said in all this that so long as imaginary quantities were still based on a fiction, they were not, so to say, fully accepted in mathematics but were regarded rather as something to be tolerated; they remained far from being given the same status as real quantities. There is no longer any justification for such discrimination now that the metaphysics of imaginary numbers has been put in a true light and that it has been shown that they have just as good a real objective meaning as the negative numbers". The authors of [29] continue with the words: "It was the authority of Gauss that first removed from complex numbers all aura of mysticism: his simple interpretation of complex numbers as points in the plane freed these fictive magnitudes from all mysterious and speculative associations and gave them full citizenship rights in mathematics as those enjoyed by the real numbers".

With respect to later developments, the same authors refer to the following: "In his habilitation presentation at Göttingen at which C.F. Gauss was present ..., R. Dedekind said ... 'Until now we have had available no theory of complex numbers entirely free from reproach...or at least none has so far been published [31]' ".

In [32], M. J. Crowe writes that C.F. Gauss published in 1831 "the geometrical justification of complex numbers, which he had worked out in 1799. Whereas ... "other"... authors on this subject attracted almost no attention, the prestige and proven track record of Gauss ensures the widespread acceptance of this representation followed upon his publication. Ironically, Gauss himself did not accept the geometrical justification of imaginaries as fully satisfactory".

Regardless of the last two statements, today's readers of a text on complex numbers do not bother too much with the question of whether $i = \sqrt{-1}$ has a meaning or what it consists of. However, they should do so for understanding how to close the historical gap in mathematical rigor which nevertheless still exists in the extensive literature including the internet. Many readers will accept the juxtaposed equations $(x, y)^2 = (x^2 - y^2, 2xy)$, $i = (0, 1)$ and $i^2 = -1$ when done in a text about complex numbers, but would surely not accept it when done in any other type of text because of an apparent conflict.

The authors of [29] also point out that A.L. Cauchy said in [33], "We call an imaginary expression, any symbolic expression of the form

$$a + b\sqrt{-1} \text{ where } a, b \text{ denote two real quantities}\ldots \tag{1}$$

Every imaginary equation is only just the symbolic representation of two equations between real quantities." A.L. Cauchy also undertakes to explain what a formal expression of the form (1) is. H. Hankel called Cauchy's explanation in this regard a conjuring trick, see [34]. The reader is encouraged to reconsider also Cauchy's other mentioned point of view, which is rarely if ever encountered these days.

Let $\mathbb{C} = (\mathbb{R}^2, \oplus, \circledast, e, i)$ where $\begin{pmatrix} x_1 \\ y_1 \end{pmatrix} \oplus \begin{pmatrix} x_2 \\ y_2 \end{pmatrix} = \begin{pmatrix} x_1 + x_2 \\ y_1 + y_2 \end{pmatrix}$ denotes usual vector addition, $e = \begin{pmatrix} 1 \\ 0 \end{pmatrix}, i = \begin{pmatrix} 0 \\ 1 \end{pmatrix}$ are particular (ordered pairs of reals or) vectors, a vector-valued vector multiplication is defined by

$$z_1 \circledast z_2 = \begin{pmatrix} x_1 x_2 - y_1 y_2 \\ x_1 y_2 + x_2 y_1 \end{pmatrix} \text{ for all } z_1 = \begin{pmatrix} x_1 \\ y_1 \end{pmatrix}, z_2 = \begin{pmatrix} x_2 \\ y_2 \end{pmatrix} \text{ from } \mathbb{R}^2, \tag{2}$$

and $+$ and $\cdot$ denote addition and multiplication of real numbers. Then, in $\mathbb{C}$, the laws of commutativity, associativity and distributivity apply, so that $\mathbb{C}$ is a field that is usually called the complex plane.

In the literature, a preliminary step on a possible way to introduce the product (2) arises if one assumes that for operating with two numbers $z_1 = \begin{pmatrix} x_1 \\ y_1 \end{pmatrix} = x_1 e + y_1 i$ and $z_2 = \begin{pmatrix} x_2 \\ y_2 \end{pmatrix} = x_2 e + y_2 i$ the usual bracket rules apply. Then, as W.R. Hamilton derived in [35],

$$\begin{pmatrix} x_1 \\ y_1 \end{pmatrix} \begin{pmatrix} x_2 \\ y_2 \end{pmatrix} = x_1 x_2 e + (x_1 y_2 + x_2 y_1) i + y_1 y_2 i^2. \tag{3}$$

For making this definition complete, he assumes that the product rule is satisfied, stating that the length of the product $z$ of $z_1$ and $z_2$ is equal to the product of the lengths of $z_1$ and $z_2$ where the length of a complex number $z = xe + yi$ is defined as $|z| = \sqrt{x^2 + y^2}$. Then,

$$i^2 = -e, \tag{4}$$

Equation (3) necessarily becomes (2), and the distributivity rule turns out to be a consequence of the other assumptions.

It is well known that $\mathbb{C}$ is a field extension of $(\mathbb{R}, +, \cdot)$ and as such is uniquely determined up to isomorphism. One may ask whether or not this uniqueness statement is preserved if one somehow weakens the assumed field properties. Since, as W.R. Hamilton showed, distributivity can follow from other properties, one could consider not requiring it from the start.

In [36], B. Riemann points out another aspect of introducing complex numbers, as stated in [29]: "In his 1851 Göttingen inaugural dissertation …'The original purpose and immediate objective in introducing complex numbers into mathematics is to express laws of dependence between variables by simpler operations on the quantities involved…,' ".

The special case of introducing complex numbers contained in [9] for the case $p = 2$ ensures compliance with the mathematical rigor reminded of above. The gap in mathematical precision described has thus been closed. In the case of arbitrary real $p > 0$, this work dispenses with distributivity of the number system under consideration and demonstrates the newly emerging possibility of constructing complex algebraic structures of great diversity.

How to go from a quadratic equation that cannot be solved for real numbers to a system of two equations that can be solved for complex numbers is shown in [18]. It is also demonstrated there that the variety of possible number systems can be further increased, and how. All of these number systems are suitable for expressing different dependencies between two variables, such as electric current and voltage, to name just one particularly striking representative.

The well known formula which deals with complex numbers in the Euclidean unit circle and is named after L.Euler [37] establishes a connection between this circle and the imaginary unit. This formula has been modified in [9,18] for several number systems, revealing different properties of the imaginary unit as an element of different spaces.

The paper is further organized as follows. A complex algebraic structure which is related to the positively homogeneous Minkowski functional $||.||$ of a bounded star body is introduced in Section 2 and a corresponding Euler type formula is derived in Section 3. The notion of generalized complex differentiability and correspondingly generalized Cauchy–Riemann differential equations are presented in Section 4. Section 5 deals with random generalized complex numbers and their basic properties, such as moments, uniform probability distributions on discs with respect to the functional $||.||$ and the circles that bound them. In addition, a connection to the equation

$$\{e^{i_p x}, x \in [0, 2\pi)\} = \left\{ \begin{pmatrix} x \\ y \end{pmatrix} \in \mathbb{R}^2 : |x|^p + |y|^p = 1 \right\} = S_p, i_p \in \mathbb{C}_p, p > 0 \tag{5}$$

following from [9] is established using generalized circle numbers from [38], and star-shaped distributed generalized complex numbers are introduced. The paper is finished by a discussion in Section 5 and Appendix A demonstrating existence and variety of generalized complex numbers.

Why do we continue to speak of random complex numbers instead of complex random numbers? We do this because the introduction of complex algebraic structures in what follows must be given much greater weight than the subsequent formal simple step of introducing randomness.

## 2. The Complex Algebraic Structure

In this section and in the Appendix A we discuss the existence and variety of complex numbers and their generalizations. Let $V$ be a two-dimensional real vector space with vector addition $\oplus$ and scalar multiplication $\cdot$ and denote by $\mathfrak{o}$ the additive neutral element. Suppose further that $||.|| : V \to \mathbb{R}^+$ is a **p**ositively **h**omogeneous and **b**ounded functional such that the set $B = \{\mathfrak{x} \in V : ||\mathfrak{x}|| \leq 1\}$ is **s**tar-shaped with respect to the inner point $\mathfrak{o}$. For the sake of brevity we call such functional phbs-functional, $B$ the unit disc with respect to the functional $||.||$ and the boundary $S = \partial B = \{\mathfrak{x} \in V : ||\mathfrak{x}|| = 1\}$ the corresponding unit circle.

**Definition 1.** *A commutative function $\odot : V \times V \to V$ is called a vector-valued vector product with respect to a phbs-functional $||.||$, or a phbs-generated vector product, if for all $\mathfrak{x}$ and $\mathfrak{y}$ from $V$ and positive reals $\lambda$ and $\mu$*

$$\mathfrak{x} \odot \mathfrak{y} = \mathfrak{o} \text{ if and only if } \mathfrak{x} = \mathfrak{o} \text{ or } \mathfrak{y} = \mathfrak{o}, \tag{6}$$

$$\frac{\mathfrak{x}}{||\mathfrak{x}||} \odot \frac{\mathfrak{y}}{||\mathfrak{y}||} \in S \text{ if } \mathfrak{x} \neq \mathfrak{o} \text{ and } \mathfrak{y} \neq \mathfrak{o} \tag{7}$$

*and*

$$(\lambda \mathfrak{x}) \odot (\mu \mathfrak{y}) = (\lambda \mu) \mathfrak{x} \odot \mathfrak{y}. \tag{8}$$

**Definition 2.** *If there exist linearly independent elements $e$ and $i$ from $V$ such that*

$$e \odot \mathfrak{x} = \mathfrak{x} \text{ for all } \mathfrak{x} \in V \tag{9}$$

*and*

$$i \odot i = -e \tag{10}$$

*then $\mathbb{V} = (V, \oplus, \odot, \cdot, \mathfrak{o}, e, i)$ is called a two-dimensional (phbs-functional related) complex algebraic structure and the elements of $V$ are called (phbs-functional related) generalized complex numbers.*

If $(V, \odot) = (\mathbb{R}^2, \odot_p)$ according to [9] then we write $\mathbb{V} = \mathbb{C}_p$ and if $(V, \odot) = (\mathbb{R}^2, \odot_{||.||})$ according to [18] then we write $\mathbb{V} = \mathbb{C}_{||.||}$.

Note that (7) entails that

$$||\mathfrak{x} \odot \mathfrak{y}|| = ||\mathfrak{x}|| \cdot ||\mathfrak{y}||. \tag{11}$$

The sets

$$B_\varrho = \varrho B = \{\varrho \mathfrak{x} : \mathfrak{x} \in B\} \text{ and } S_\varrho = \varrho S$$

being generated by the phbs-functional $||.||$ are correspondingly called disc and circle having phbs-radius $\varrho > 0$ and center $\mathfrak{o}$. By

$$z_1 \odot z_2 = ||z_1|| ||z_2|| \frac{z_1 \circledast z_2}{||z_1 \circledast z_2||} \tag{12}$$

we are always given a vector-valued vector product with respect to a phbs-functional $||.||$.

**Definition 3.** *The operation $z_1 \odot z_2 : V \times V \to V$ defined by (12) will be called generalized complex multiplication (with respect to the phbs-functional $||.|| : V \to \mathbb{R}^+$).*

To finish this section, we recall that if $V = \mathbb{R}^2$ and $\odot = \circledast$ then $\mathbb{V}$ is usually called the complex plane. For other choices of $V$, $\mathbb{V}$ is called the *V*-realization of the complex plane.

### 3. Euler-Type Trigonometric Representation of Generalized Complex Numbers

From now on, for the sake of simplicity, let $V = \mathbb{R}^2$ and assume that a vector-valued vector product with respect to a phbs-functional is given according to Definition 1.

**Definition 4.** *With $z^0 = \frac{e}{||e||}, z^1 = z$, the n-th vector-valued power of $z \in V$ is defined as*

$$z^n = z^{n-1} \odot z, n = 2, 3, \dots$$

**Definition 5.** *The vector-valued exponential function with respect to the phbs-funtional $||.||$, $\exp_{||.||}(.) : V \to V$, is defined as*

$$\exp_{||.||}(z) = \sum_{k=0}^{\infty} \frac{z^k}{k!}.$$

**Definition 6.** *Let the central projection of vector $z \in V$ onto the unit circle $S$ be denoted $cpr_S(z)$. The exponential-projection function with respect to the phbs-functional $||.||$, $e_{||.||} : V \to S$, is defined then by*

$$e_{||.||}^z = cpr_S(\exp_{||.||}(z)).$$

**Definition 7.** *The generalizations of the usual cosine and sine functions with respect to the phbs-funtional $||.||$ are defined to be*

$$\cos_S x = \frac{\cos x}{N(x)} \text{ and } \sin_S x = \frac{\sin x}{N(x)}$$

*where*

$$N(x) = || \cos x \frac{e}{||e||} + \sin x \frac{i}{||i||} ||.$$

Note that

$$\partial B = \{(\cos_S \varphi)e + (\sin_S \varphi)i, 0 \leq \varphi < 2\pi\}.$$

In the following theorem, it is assumed that there is exactly one fixed imaginary unit for all phbs-functionals. However, the right-hand side of the Euler-type formula explicitly depends on the specific choice of the phbs-functional in two ways, namely explicitly by $||e||$ and $||i||$ and implicitly through $S$ in the definition of the generalized sine and cosine functions. In the subsequent Corollary it is conversely assumed that the imaginary unit is chosen in dependence of the phbs-functional. At first glance, the right-hand side of the Euler-type formula then seems to depend only implicitly through $S$ from the choice of the phbs-functional because the dependence of the imaginary unit on the phbs-functional is not visible there.

**Theorem 1.** *If*

$$e = \begin{pmatrix} 1 \\ 0 \end{pmatrix} \text{ and } i = \begin{pmatrix} 0 \\ 1 \end{pmatrix} \tag{13}$$

*then the following Euler-type formula holds true*

$$e_{||.||}^{x \frac{i}{||i||}} = \cos_S x \frac{e}{||e||} + \sin_S x \frac{i}{||i||}, \; x \in \mathbb{R}. \tag{14}$$

**Proof.** It follows from

$$i^{2k} = (-1)^k ||i||^{2k} \frac{e}{||e||} \text{ and } i^{2k+1} = (-1)^k ||i||^{2k} i$$

that

$$e_{||\cdot||}^{x\frac{i}{||i||}} = cpr_S(\exp_{||\cdot||}(x\frac{i}{||i||})) = \frac{\cos x \frac{e}{||e||} + \sin x \frac{i}{||i||}}{N(x)}.$$

□

**Remark 1.** *We emphasize that (14) is a vector equation. While one-dimensional considerations in the literatur have so far not led to a fully satisfactory explanation for the imaginary unit, the two-dimensional consideration carried out here allows different interpretations in a well-structured way, as will be shown below.*

With $e$ and $i$ as in this theorem, for any phbs-functional $||\cdot||$, the following vector equation holds,

$$e_{||\cdot||}^{\pi\frac{i}{||i||}} + \frac{e}{||e||} = o. \tag{15}$$

**Corollary 1.** *If*

$$e = \frac{\begin{pmatrix} 1 \\ 0 \end{pmatrix}}{||\begin{pmatrix} 1 \\ 0 \end{pmatrix}||} \text{ and } i = \frac{\begin{pmatrix} 0 \\ 1 \end{pmatrix}}{||\begin{pmatrix} 0 \\ 1 \end{pmatrix}||} \tag{16}$$

*then we have the following Euler-type formula*

$$e_{||\cdot||}^{xi} = (\cos_S x)e + (\sin_S x)i. \tag{17}$$

**Proof.** Obviously,

$$||e|| = ||i|| = 1 \text{ and } N(x) = ||(\cos x)e + (\sin x)i||.$$

Thus,

$$i^{2k} = (-1)^k e \text{ and } i^{2k+1} = (-1)^k i$$

and

$$e_{||\cdot||}^{xi} = cpr_S(\exp_{||\cdot||}(xi)) = \frac{\cos xe + \sin xi}{N(x)}.$$

□

With $e$ and $i$ as in this corollary, the following vector equation is true for any phbs-functional $||\cdot||$,

$$e_{||\cdot||}^{i\pi} + e = o. \tag{18}$$

This equation clarifies and generalizes the well known formula

$$e^{i\pi} + 1 = 0. \tag{19}$$

**Remark 2.** *If $||\cdot|| = ||\cdot||_p$ where $||z||_p = (|x|^p + |y|^p)^{1/p}$ then in Corollary 1 there holds $i = \begin{pmatrix} 0 \\ 1 \end{pmatrix}, \forall p > 0$, but $e_{||\cdot||_p}^{xi}$ nevertheless depends on $p$.*

**Corollary 2.** *Let $z = \begin{pmatrix} x \\ y \end{pmatrix}$ from V and assumtion (16) be fulfilled. Then z allows the generalized polar representation*

$$z = re^{i\varphi}_{||.||} \tag{20}$$

*with*

$$r = ||\begin{pmatrix} x \\ y \end{pmatrix}|| \text{ and } \arctan|\frac{y}{x}| = \begin{cases} \varphi & \text{if } z \in Q1 \\ \pi - \varphi & \text{if } z \in Q2 \\ \varphi - \pi & \text{if } z \in Q3 \\ 2\pi - \varphi & \text{if } z \in Q4 \end{cases} \tag{21}$$

*where $Q1 - Q4$ stand for the anticlockwise enumerated quadrants of $\mathbb{R}^2$.*

**Proof.** The generalized polar coordinate transformation w.r.t the phbs-functional $||.||$, $Pol_S : [0, 2\pi) \times [0, \infty) \rightarrow \mathbb{R}^2$, is defined in [38] by $x = r\cos_S \varphi, y = r\sin_S \varphi$ that is $z = r((\cos_S \varphi)e + (\sin_S \varphi)i)$. The inverse of $Pol_S$ is given by (20), (21). Finally, (17) applies to prove (20). □

For $i$ satisfying (16), let us write $i = i_{||.||}$. We close this section with the following generalization of Equation (5):

$$\{e^{i_{||.||}x}_{||.||}, x \in [0, 2\pi)\} = \{\begin{pmatrix} x \\ y \end{pmatrix} \in \mathbb{R}^2 : ||\begin{pmatrix} x \\ y \end{pmatrix}|| = 1\} = S_{||.||}, i_{||.||} \in \mathbb{C}_{||.||}. \tag{22}$$

## 4. Generalized Complex Differentiability

We note that both the usual complex multiplication

$$z_1 \circledast z_2 = \begin{pmatrix} x_2 & -y_2 \\ y_2 & x_2 \end{pmatrix} \begin{pmatrix} x_1 \\ y_1 \end{pmatrix}$$

and the generalized complex multiplication (12) represent torsional stretches and that

$$\begin{pmatrix} x & -y \\ y & x \end{pmatrix} \begin{pmatrix} x & y \\ -y & x \end{pmatrix} = ||z||^2 \begin{pmatrix} 1 & 0 \\ 0 & 1 \end{pmatrix}, z = \begin{pmatrix} x \\ y \end{pmatrix},$$

which motivates the following definition.

**Definition 8.** *With*

$$z_1 \oslash z_2 = \begin{pmatrix} x_2 & y_2 \\ -y_2 & x_2 \end{pmatrix} \begin{pmatrix} x_1 \\ y_1 \end{pmatrix} = \begin{pmatrix} x_1 x_2 + y_1 y_2 \\ y_1 x_2 - x_1 y_2 \end{pmatrix},$$

*the generalized complex division (with respect to the phbs-functional $||.||$) is defined as*

$$z_1 \oslash_{||.||} z_2 = \frac{||z_1||}{||z_2||} \frac{z_1 \oslash z_2}{||z_1 \oslash z_2||}.$$

It can be easily checked that

$$(z_1 \oslash_{||.||} z_2) \odot z_2 = z_1$$

and, for $\varepsilon \in \mathbb{R}$,

$$(\varepsilon z) \oslash_{||.||} \begin{pmatrix} \varepsilon \\ 0 \end{pmatrix} = \frac{z}{||e||}$$

as well as

$$(\varepsilon z) \oslash_{||.||} \begin{pmatrix} 0 \\ \varepsilon \end{pmatrix} = \frac{||z||}{||i|| \cdot ||Az||} Az \text{ where } A = \begin{pmatrix} 0 & 1 \\ -1 & 0 \end{pmatrix}$$

describes a 90° counterclockwise rotation.

**Definition 9.** *Let U be an open subset of the complex plane $\mathbb{C}$. A function $f = \begin{pmatrix} u \\ v \end{pmatrix} : U \to \mathbb{C}$ is called differentiable at $z_0 \in \mathbb{C}$ if the limit*

$$\lim_{z \to z_o} (f(z) - f(z_o)) \oslash_{||\cdot||} (z - z_o)$$

*exists. Moreover, f is called a holomorphic function if it is generalized complex differentiable in U.*

**Theorem 2.** *If $f = \begin{pmatrix} u \\ v \end{pmatrix}$ is differentiable in $U \subset \mathbb{C}$ then the partial derivatives of u and v satisfy the equations*

$$u_x(x,y) = Q(x,y)v_y(x,y), \quad v_x(x,y) = -Q(x,y)u_y(x,y) \tag{23}$$

*where $Q(x,y) = \dfrac{||e|| \cdot || \begin{pmatrix} u_y \\ v_y \end{pmatrix} ||}{||i|| \cdot || \begin{pmatrix} v_y \\ -u_y \end{pmatrix} ||}$.*

**Proof.** Recognize that, with some $\delta \in (0,1)$,

$$(f(z + \varepsilon e) - f(z)) \oslash_{||\cdot||} (\varepsilon e)$$

$$= (( \begin{pmatrix} u(x + \varepsilon, y) \\ v(x + \varepsilon, y) \end{pmatrix} ) - ( \begin{pmatrix} u(x,y) \\ v(x,y) \end{pmatrix} )) \oslash_{||\cdot||} \begin{pmatrix} \varepsilon \\ 0 \end{pmatrix}$$

$$= \begin{pmatrix} \varepsilon u_x(x + \delta\varepsilon, y) \\ \varepsilon v_x(x + \delta\varepsilon, y) \end{pmatrix} \oslash_{||\cdot||} \begin{pmatrix} \varepsilon \\ 0 \end{pmatrix} \tag{24}$$

$$= \frac{1}{||e||} \begin{pmatrix} u_x(x + \delta\varepsilon, y) \\ v_x(x + \delta\varepsilon, y) \end{pmatrix}$$

and

$$(f(z + \varepsilon i) - f(z)) \oslash_{||\cdot||} (\varepsilon i) = \frac{|| \begin{pmatrix} u_y(x, y + \delta\varepsilon) \\ v_y(x, y + \delta\varepsilon) \end{pmatrix} ||}{||i|| \cdot ||A \begin{pmatrix} u_y(x, y + \delta\varepsilon) \\ v_y(x, y + \delta\varepsilon) \end{pmatrix} ||} A \begin{pmatrix} u_y(x, y + \delta\varepsilon) \\ v_y(x, y + \delta\varepsilon) \end{pmatrix}.$$

Thus, for $\varepsilon \to 0$,

$$(f(z + \varepsilon e) - f(z)) \oslash_{||\cdot||} (\varepsilon e) \to \frac{1}{||e||} \begin{pmatrix} u_x(x,y) \\ v_x(x,y) \end{pmatrix}$$

and

$$(f(z + \varepsilon i) - f(z)) \oslash_{||\cdot||} (\varepsilon i) \to \frac{|| \begin{pmatrix} u_y(x,y) \\ v_y(x,y) \end{pmatrix} ||}{||i|| \cdot || \begin{pmatrix} v_y(x,y) \\ -u_y(x,y) \end{pmatrix} ||} \begin{pmatrix} v_y(x,y) \\ -u_y(x,y) \end{pmatrix}$$

from where (23) immediately follows.　□

**Definition 10.** *We call the equations in Theorem 2 the generalized Cauchy–Riemann differential equations (with respect to the phbs-funtional $||\cdot||$).*

## 5. Random Numbers

### 5.1. Moments

Let $[\Omega, \mathfrak{A}, P]$ denote any probability space and $\mathfrak{S}$ the Borel $\sigma$-algebra of subsets of $V$. Any $(\mathfrak{A}, \mathfrak{S})$-measurable mapping $Z = \begin{pmatrix} X \\ Y \end{pmatrix} : \Omega \to V$ will be called a random generalized complex number. If we assume that the square $z^2 = z \odot z$ of a complex number $z$ is particularly defined as $z^2 = z \circledast z$, $Z$ possesses second order moments and $\mathbb{E}$ means mathematical expectation then the vector-valued second order central moment of $Z$ is

$$\mathbb{E}(Z - \mathbb{E}Z)^2 = \begin{pmatrix} \sigma_X^2 - \sigma_Y^2 \\ 2\sigma_{X,Y} \end{pmatrix},$$

where $\sigma_X^2$ and $\sigma_Y^2$ stand accordingly for the variances of $X$ and $Y$ and $\sigma_{X,Y}$ for the covariance between $X$ and $Y$. If $||.|| = ||.||_2$ denotes the Euclidean norm then $\tau = ||\mathbb{E}(Z - \mathbb{E}Z)^2||_2^2$ takes the form

$$\tau = (\sigma_X^2 - \sigma_Y^2)^2 + 4\sigma_{X,Y}^2.$$

Note that $\tau = 0$ means homoscedasticity and uncorrelatedness of $X$ and $Y$, and non-zero $\tau$ means either heteroscedasticity or correlatedness, or both.

Similarly, the vector-valued third and fourth order central moments of $Z$ are, under suitable conditions,

$$\mathbb{E}(Z - \mathbb{E}Z)^3 = \begin{pmatrix} \mathbb{E}(X - \mu_X)^3 - 3\mathbb{E}(X - \mu_X)(Y - \mu_Y)^2 \\ 3\mathbb{E}(X - \mu_X)^2(Y - \mu_Y) - \mathbb{E}(Y - \mu_y)^3 \end{pmatrix}$$

and

$$\mathbb{E}(Z - \mathbb{E}Z)^4 = \begin{pmatrix} \mathbb{E}(X - \mu_X)^4 - 6\mathbb{E}(X - \mu_X)^2(Y - \mu_Y)^2 + \mathbb{E}(Y - \mu_Y)^4 \\ 4\mathbb{E}(X - \mu_X)^3(Y - \mu_Y) - 4\mathbb{E}(X - \mu_X)(Y - \mu_Y)^3 \end{pmatrix},$$

respectively.

### 5.2. Uniform Probability Distribution

A closer description of a random variable is given by its distribution law. We start with one of the most elementary distributions which one can assign a random vector in $V$. If $\mu(A)$ denotes the area content or Lebesgue measure of $A$, $\mu(A) = \int_A d(x,y)$, where $A$ is a Borel subset of $V$, then the uniform probability distribution on $B \in \mathfrak{S}$ is defined by

$$\mathfrak{U}(A) = \frac{\mu(A)}{\mu(B)}, A \in \mathfrak{S} \cap B.$$

Let us assume that $Z$ follows this distribution, $Z \overset{d}{\sim} \mathfrak{U}$. The probability density of the random vector $Z$ is then

$$f_Z(x,y) = \frac{1}{\mu(B)}, (x,y)^T \in B.$$

We now exploit the fact that $Z$ in the sense of Corollary 2 allows the generalized polar representation

$$Z = Pol_S(R, \Phi) = R\begin{pmatrix} \cos_S \Phi \\ \sin_S \Phi \end{pmatrix} = Re_{||.||}^{i\Phi}. \tag{25}$$

The Jacobian of the transformation $Pol_S$ is $\frac{D(x,y)}{D(r,\varphi)} = \frac{r}{N^2(\varphi)}$, the area content of $B_\varrho$ satisfies the equation

$$\mu(B_\varrho) = \mu(B)\varrho^2 \text{ where } \mu(B) = \frac{1}{2} \int\limits_0^{2\pi} \frac{d\varphi}{N^2(\varphi)}, \tag{26}$$

and the joint distribution function of $R$ and $\Phi$ is given by

$$P(R < \varrho, \Phi < \psi) = \int\limits_0^\varrho \int\limits_0^\psi \frac{1}{\mu(B)} \frac{r}{N^2(\varphi)} d\varphi dr, \varrho > 0, \psi \in (0, 2\pi).$$

Thus, $R$ and $\Phi$ are stochastically independent and follow the densities

$$f_R(r) = 2r, 0 < r < 1 \tag{27}$$

and

$$f_\Phi(\varphi) = \frac{1}{2\mu(B)N^2(\varphi)}, \varphi \in (0, 2\pi), \tag{28}$$

respectively. This allows to simulate $R, \Phi$ and $Z$.

*5.3. Generalized Circle Numbers*

In this section, we discuss a geometric aspect of the normalizing constant $\mu(B)$ from (28). The phbs-functional $||.||$ can be viewed as the Minkowski functional of the star body $B$, that is $||z|| = h_B(z) = \inf\{\lambda > 0 : z \in \lambda B\}, z \in V$.

Let $T$ be a star disc in $\mathbb{R}^2$ which is generated by the positive homogeneous Minkowski functional $h_T$ and $\mathfrak{Z}_n : z_0, z_1, ..., z_n = z_0$ a successive and positive (anticlockwise) oriented partition of $S_\varrho$. The positive directed $T$-arc-length of $S_\varrho$ is defined as

$$AL_{S_\varrho,T} = \lim_{n \to \infty} \sum_{j=1}^\infty h_T(z_j - z_{j-1}), \tag{29}$$

if the limit exists for and is independent of all described partitions of $S_\varrho$ satisfying the assumption $\max_{1 \le j \le n} h_T(z_j - z_{j-1}) \to 0$ as $n \to \infty$.

If the gradient $\nabla h_B$ is defined almost everywhere and a star body $S^*$ satisfies the rotated gradient condition

$$h_{\begin{pmatrix} 0 & 1 \\ -1 & 0 \end{pmatrix} S^*} (\nabla h_B(x,y)|_{(x,y)=Pol_s(r,\varphi)}) = 1, a.e. \tag{30}$$

Then, according to [38], $AL_{S_\varrho,S^*}$ allows the representation,

$$AL_{S_\varrho,S^*} = 2\mu(B)\varrho. \tag{31}$$

It follows from (26) and (31) that

$$\frac{\mu(B_\varrho)}{\varrho^2} = \mu(B) = \frac{AL_{S_\varrho,S^*}}{2\varrho},$$

which was the motivation in [38] to call

$$\pi_S := \mu(B) \tag{32}$$

the $S$-generalized circle number of the star body $B$. With regard to the specific definition of $S^*$, we refer to [38,39] for the general case.

We note that, if specifically $V = \mathbb{R}^2$ and $\left|\left|\begin{pmatrix} x \\ y \end{pmatrix},\right|\right| = \sqrt{x^2 + y^2}$ denotes the Euclidean norm then the quantity $\pi_S = \pi$ is well known to represent the area content of the unit disc, and $S = S^*$, $\mu(B) = \pi$.

If, however, $\left|\left|\begin{pmatrix} x \\ y \end{pmatrix}\right|\right| = \sqrt{(\frac{x}{a})^2 + (\frac{y}{b})^2}$ then $B = \{\begin{pmatrix} x \\ y \end{pmatrix} : \frac{x^2}{a^2} + \frac{y^2}{b^2} \leq 1\}$ has area content $\mu(B) = ab\pi$, $\pi_S = ab\pi$, and $S^* = \{\begin{pmatrix} x \\ y \end{pmatrix} : b^2x^2 + a^2y^2 = 1\}$ satisfies (30).

*5.4. The Distribution of $e_{||\cdot||}^{i\Phi}$*

Let $\Phi$ still denote the random angle from the stochastic representation (25) and assume for the sake of simplicity from now on that assumption (16) is satisfied. The random generalized complex number $e_{||\cdot||}^{i\Phi}$ takes values in $S$, $e_{||\cdot||}^{i\Phi} : \Omega \to S$, and is $(\mathfrak{A}, \mathfrak{B}(S))$-measurable where $\mathfrak{B}(S)$ denotes the Borel $\sigma$-algebra over $S$. The following sections of $S$,

$$S(\varphi_1, \varphi_2) = \{\begin{pmatrix} \cos_S \varphi \\ \sin_S \varphi \end{pmatrix} : \varphi_1 \leq \varphi \leq \varphi_2\}, 0 \leq \varphi_1 < \varphi_2 < 2\pi,$$

are elements from $\mathfrak{B}(S)$ and serve for defining the following sectors of $B$:

$$sector(\varphi_1, \varphi_2) = \{r \cdot S(\varphi_1, \varphi_2), 0 \leq r < 1\}, 0 \leq \varphi_1 < \varphi_2 < 2\pi.$$

In analogy to formulae (2.35) and (2.36) in [38], we have

$$AL_{\varrho \cdot S(\varphi_1, \varphi_2), S^*} = \varrho \int_{\varphi_1}^{\varphi_2} \frac{d\varphi}{N^2(\varphi)} \tag{33}$$

and

$$\mu(\varrho \cdot sector(\varphi_1, \varphi_2)) = \frac{\varrho^2}{2} \int_{\varphi_1}^{\varphi_2} \frac{d\varphi}{N^2(\varphi)}. \tag{34}$$

Notice that (33) and (34) together broadly generalize the defining equations of $\pi_S$ and are an integral part of the intellectual background of the generalized Cavalieri integration mentioned in [40]. Moreover,

$$P(e_{||\cdot||}^{i\Phi} \in S(\varphi_1, \varphi_2)) = P(\begin{pmatrix} \cos_S(\Phi) \\ \sin_S(\Phi) \end{pmatrix} \in S(\varphi_1, \varphi_2)) = P(\Phi \in (\varphi_1, \varphi_2)).$$

It follows now by (28) and (31) that the distribution law $\mathfrak{L}(e^{i\Phi})$ looked for in this section is

$$P(e^{i\Phi} \in S(\varphi_1, \varphi_2)) = \frac{AL_{S(\varphi_1, \varphi_2), S^*}}{AL_{S, S^*}}, 0 \leq \varphi_1 < \varphi_2 < 2\pi. \tag{35}$$

**Definition 11.** *We call the distribution in (35) the generalized uniform distribution on the circle $S$ with respect to the phbs-functional $||.||$, or the $S$-generalized uniform distribution, for short, and a random variable $\mathcal{U}_S$ that follows this distribution an $S$-generalized uniform basis of the random generalized complex number $Z$.*

Equation (25) can now be viewed as a stochastic representation,

$$Z \sim R \cdot \mathcal{U}_S, \tag{36}$$

where the random phbs-radius variable $R = ||Z||$ and uniform basis $\mathcal{U}_S$ are independent and follow the density (27) and the distribution law (35), respectively.

Because $S$ borders the star body $B$, the product $R \cdot \mathcal{U}_S$ is said to be star-shaped distributed.

*5.5. Star-Shaped Distributed Random Generalized Complex Numbers*

**Definition 12.** *A random generalized complex number* $Z = \begin{pmatrix} X \\ Y \end{pmatrix}$ *is said to follow a star-shaped distribution if there exist a nonnegative random variable R, a star body S, centered at* $\mathfrak{o}$*, and a uniform basis* $\mathcal{U}_S$ *that is independent of R such that the random vector* $\begin{pmatrix} X \\ Y \end{pmatrix}$ *satisfies the stochastic representation (36).*

For the big subclasses of norm- and antinorm-contoured and more general star-shaped distributions, we refer to [40,41], respectively. For antinorms, see [42]. The more particular class of elliptically contoured distributions is studied in [12,40,41,43] and the related ellipses numbers in [38,39].

Still another class of probability densities which are invariant with respect to certain $[p, q]$-vector multiplications was discussed in [18].

## 6. Discussion

In the present work, a new, rather general concept of complex numbers was brought into connection with the concepts of generalized uniform distributions on generalized circles, generalized circle numbers for circular discs with respect to positively homogeneous and bounded functionals and a well known general theory of star-shaped distributions in $\mathbb{R}^2$. Moreover, the concept of complex differentiability was generalized and a generalization of the Cauchy–Riemann differential equations was derived. Obvious further questions for future work in this area concern a general treatment of power series and the search for a potentially possible adaption of the statement of Cauchy's integral theorem. Additionally, the reader should be made aware of the following aspects.

As subsets of the real line, the set of all integers is a subset of the set of all rational numbers; analogously, every rational number is a real number. However, are real numbers special complex numbers? To the best of the author's knowledge, this has actually not been shown anywhere, although an impression to this effect may occasionally have arisen. Instead, it has been proved that the two-dimensional field of complex numbers, $\mathbb{C}$, is an extension field of the one-dimensional field of real numbers, $\mathbb{R}$, based on the following. Because of

$$\begin{pmatrix} x_1 \\ 0 \end{pmatrix} \oplus \begin{pmatrix} x_2 \\ 0 \end{pmatrix} = \begin{pmatrix} x_1 + x_2 \\ 0 \end{pmatrix} \text{ and } \begin{pmatrix} x_1 \\ 0 \end{pmatrix} \odot \begin{pmatrix} x_2 \\ 0 \end{pmatrix} = \begin{pmatrix} x_1 \cdot x_2 \\ 0 \end{pmatrix},$$

the mapping $x \mapsto \begin{pmatrix} x \\ 0 \end{pmatrix}$ defines an isomorphism from $\mathbb{R}$ to $\mathbb{C}$. In other words, the space $(\{\begin{pmatrix} x \\ 0 \end{pmatrix}, x \in \mathbb{R}\}, \oplus, \odot)$ is isomorph to $(\mathbb{R}, +, \cdot)$. The occasionally encountered view that $z = x + iy = \begin{pmatrix} x \\ y \end{pmatrix}$ becomes a real number if $y = 0$ falls short within the present vector space approach.

In a certain part of the international mathematical literature, a mentality of wishing properties of abstract mathematical objects has become established without giving the reader the guarantee that these wishes can be fulfilled, or how, by explicitly specifying suitable concrete mathematical objects. Beginning in [9,18], this approach was abandoned and explicit objects for the realization of ordinary complex numbers and their $p$-generalizations, $p > 0$, as well their norm-, antinorm- and semi-antinorm-generalizations were given.

As we have seen, Equation (19), often referred to as mysterious or most beautiful formula of mathematics, does not, from a rigorous point of view, point to the facts at hand with complete precision. The message that a frequently quoted formula is not perfectly correct could lead to great uncertainty, if it had not been specified in Equation (18) at the same time.

These circumstances are the reason why we also subject other well-known mathematical formulas to a check for the necessary mathematical rigor. This is demonstrated here using the example of the addition property of the exponential function. The proof of the formula,

$$e^{x+y} = e^x e^y \text{ for all } x \text{ and } y \text{ from } \mathbb{R},\tag{37}$$

can be done quite easily using the Cauchy formula as follows:

$$e^x e^y = \left(\sum_{k=0}^{\infty}\frac{x^k}{k!}\right)\left(\sum_{l=0}^{\infty}\frac{y^l}{l!}\right) = \sum_{m=0}^{\infty}\left(\sum_{\nu=0}^{m}\frac{x^\nu y^{m-\nu}}{\nu!(m-\nu)!}\right) = \sum_{m=0}^{\infty}\frac{(x+y)^m}{m!} = e^{x+y}.$$

However, this proof cannot be transferred directly to complex numbers within the framework of the present vector space approach. Let $||.||$ be the Euclidean norm in $\mathbb{R}^2$. In order to prove the formula

$$exp_{||.||}(z_1 + z_2) = exp_{||.||}(z_1) \circledast exp_{||.||}(z_2) \text{ for all } z_1 \text{ and } z_2 \text{ from } \mathbb{C}_2\tag{38}$$

in the vector approach considered here for ordinary complex numbers, the formulation of the series expansion

$$\begin{aligned}
exp_{||.||}(z) &= e + \begin{pmatrix} x \\ y \end{pmatrix} + \frac{1}{2!}\begin{pmatrix} x^2 - y^2 \\ 2xy \end{pmatrix} + \frac{1}{3!}\begin{pmatrix} x^3 - 3xy^2 \\ 3x^2y - y^3 \end{pmatrix} \\
&+ \frac{1}{4!}\begin{pmatrix} x^4 - 6x^2y^2 + y^4 \\ 4(x^3y - xy^3) \end{pmatrix} + \frac{1}{5!}\begin{pmatrix} x^5 - 10x^3y^2 + 5xy^4 \\ 5x^4y - 10x^2y^3 + y^5 \end{pmatrix} \\
&+ \frac{1}{6!}\begin{pmatrix} x^6 - 15x^4y^2 + 15x^2y^4 - y^5 \\ 6x^5y - 20x^3y^3 + 6xy^5 \end{pmatrix} + \dots, \ z = z_1 + z_2 \in \mathbb{C}_2
\end{aligned}\tag{39}$$

and the suitable sorting of the summands in the corresponding vector product,

$$exp_{||.||}(z_1) \circledast exp_{||.||}(z_2) \text{ for all } z_1 \text{ and } z_2 \text{ from } \mathbb{C}_2,\tag{40}$$

are correspondingly more technically challenging. The reader should wonder from the considerations made so far whether in the present approach, or other ones in papers which start from various desires that a complex algebraic structure should satisfy, there is a guarantee that all precisifications in the sense of mathematical rigor cannot lead to any technical or even deep-seated conflicts. The need for a consistent, strict mathematical action in this field is additionally underlined by a statement in [44], where it is shown in Theorem 1, Formula (19), that the exponential function considered there does not have the addition property examined here.

If one has finally carried out the steps just indicated, then, for example, a valid proof follows from (38) and the usual definitions of $\cos z$ and $\sin z$ that the Euler formula

$$e^{ix} = \cos x + i\sin x, x \in \mathbb{R}$$

can be generalized to

$$e^{iz} = \cos z + i\sin z, z \in \mathbb{C}.$$

Fortunately, the greater effort that a vector consideration of complex numbers requires is offset by a greater expected benefit. For example, it was shown in [18] that a quadratic equation that has no real solution can have solutions in $\mathbb{C}_p$ for infinitely many $p$. A fundamental question then is, for which $p$, for a given variable $x$, a suitable variable $y$ exists and is well interpretable, so that they naturally satisfy the equation $|x|^p + |y|^p = r^p$ for a certain $r > 0$.

**Funding:** This research received no external funding.

**Institutional Review Board Statement:** Not applicable.

**Informed Consent Statement:** Not applicable.

**Data Availability Statement:** Not applicable.

**Conflicts of Interest:** The author declares no conflict of interest.

## Appendix A

The examples presented in this section are to prove the existence and variety of complex numbers and their generalizations. Notice that the known uniqueness statement on usual complex numbers can only be made under the additional distributivity assumption with respect to the complex algebraic structure under consideration. Each of the following examples can occur equally in the deterministic and in the stochastic context. To name just two of the possible areas of application, we refer to complex variable frequency electric circuit theory in [45] and complex spectral signal representation for the processing and analysis of images in [46].

**Example A1.** *Let* $V = \mathbb{R}^2, ||z|| = ||\begin{pmatrix} x \\ y \end{pmatrix}|| = \sqrt{x^2 + y^2}$, $z_1 \odot z_2 = z_1 \circledast z_2$, $e = \begin{pmatrix} 1 \\ 0 \end{pmatrix}$ and $i = \begin{pmatrix} 0 \\ 1 \end{pmatrix}$. Then, the assumptions (6)–(10) are fulfilled; the complex algebraic structure $\mathbb{C} = (V, \oplus, \odot, \cdot, \mathsf{o}, e, i)$ is commonly called the complex plane and its elements, $z = \begin{pmatrix} x \\ y \end{pmatrix}$, are called complex numbers.*

*Usually complex numbers are written as $z = x + iy$ where the so called imaginary unit i is said to come from a different set than the real numbers, nowadays an astonishing non-mathematical approach. The circumstance that then it is simply not explained what $iy, x + iy, i^2$ are, and what the "identification" $-\begin{pmatrix} 1 \\ 0 \end{pmatrix} = -1$, which stands for $i^2 = \begin{pmatrix} 0 \\ 1 \end{pmatrix} \circledast \begin{pmatrix} 0 \\ 1 \end{pmatrix} = -1$ means is often not even mentioned.*

*The present approach to $\mathbb{C}$ avoids such a gap in mathematical rigor and proves the existence of a mathematically completely formally correct definition while the following examples show the variety of complex numbers as well as the existence and variety of their generalizations.*

**Example A2.** *Let $V = \{f : [0,1] \to \mathbb{R}$ with $f(x) = ax + b$ where $a, b$ are reals$\}$ be a function space, $||f|| = \sqrt{a^2 + b^2}$ a norm on it and*

$$(f \odot g)(x) = (ac - bd)x + (ad + bc)$$

*if $f(x) = ax + b$ and $g(x) = cx + d$ with reals $a, b, c, d$. Further assume that elements $e$ and $i$ from $V$ satisfy $e(x) = x$ and $i(x) = 1$ for all $x \in [0, 1]$. Then, assumptions (6)–(10) are satisfied and the complex algebraic structure $(V, \oplus, \odot, \cdot, \mathsf{o}, e, i)$ can be considered as another realization of the complex plane $\mathbb{C}$.*

**Example A3.** *Let the vector space $V = \{\begin{pmatrix} a & -b \\ b & a \end{pmatrix}, a, b$ are reals $\}$ be endowed with the norm $||\begin{pmatrix} a & -b \\ b & a \end{pmatrix}|| = \sqrt{a^2 + b^2}$ and put $e = \begin{pmatrix} 1 & 0 \\ 0 & 1 \end{pmatrix}$, $i = \begin{pmatrix} 0 & -1 \\ 1 & 0 \end{pmatrix}$ as well as*

$$\begin{pmatrix} a & -b \\ b & a \end{pmatrix} \odot \begin{pmatrix} c & -d \\ d & c \end{pmatrix} = \begin{pmatrix} ac - bd & -ad - bc \\ ad + bc & ac - bd \end{pmatrix}.$$

*Then, assumptions (6)–(10) are satisfied and the corresponding algebraic structure is called the matrix representation of $\mathbb{C}$.*

**Example A4.** *Let $V = \mathbb{R}^2$, $||z|| = ||z||_p$ with $p \neq 0$,*

$$z_1 \odot z_2 = z_1 \odot_p z_2 = \left[ \frac{(|x_1|^p + |y_1|^p)(|x_2|^p + |y_2|^p)}{|x_1 x_2 - y_1 y_2|^p + |x_1 y_2 + x_2 y_1|^p} \right]^{1/p} z_1 \circledast z_2$$

*for all $z_1 = \begin{pmatrix} x_1 \\ y_1 \end{pmatrix}, z_2 = \begin{pmatrix} x_2 \\ y_2 \end{pmatrix}$ from V and $e = \begin{pmatrix} 1 \\ 0 \end{pmatrix}, i = \begin{pmatrix} 0 \\ 1 \end{pmatrix}$. Then, the assumptions (6)–(10) are satisfied and the corresponding complex algebraic structure can be viewed as a generalization of the complex plane $\mathbb{C}$.*

*Here, $||.||_p$ denotes a norm, antinorm or semi-antinorm if accordingly $p \geq 1$, $0 < p \leq 1$ or $p < 0$, [42], and $\oplus$ denotes common vector addition.*

**Remark A1.** *Suitably modified, the product $\odot_p$ can also be used in Examples 2 and 3.*

**Example A5.** *If $V = \mathbb{R}^2$, $||z|| = |z|_{(a,b)} = \sqrt{(\frac{x}{a})^2 + (\frac{y}{b})^2}, a > 0, b > 0$ and*

$$z_1 \odot_{(a,b)} z_2 = \left( \frac{(x_1^2 + (\frac{a}{b} y_1)^2)((\frac{b}{a} x_2)^2 + y_2^2)}{b^2 (x_1 x_2 - y_1 y_2)^2 + a^2 (x_1 y_2 + x_2 y_1)^2} \right)^{1/2} z_1 \circledast z_2$$

*then assumptions (6)–(10) are satisfied and the complex algebraic structure $(V, \oplus, \odot, \cdot, \mathrm{o}, e, i)$, where $\oplus, \mathrm{o}, e, i$ are as before, is called an elliptical complex plane.*

**Remark A2.** *Suitably adopted, the Euler-type formulae apply to arbitrary two-dimensional complex algebraic structures.*

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
