# Peer review of "Deterministic and Random Generalized Complex Numbers Related to a Class of Positively Homogeneous Functionals"

_axioms, doi:10.3390/axioms12010060_

Round 1

Reviewer 1 Report

Referee’s report on

On Random Generalized Complex Numbers

by

Wolf-Dieter Richter

In this paper, Based upon a new general vector-valued vector product, generalized complex numbers with respect to certain positive homogeneous functionals including norms and antinorms are introduced  and a vector-valued Euler type formula for them is derived using a vector valued exponential function. For random such generalized complex numbers, moments are introduced and uniform distributions on discs.

 - General suggestions. Interesting ideas about generalized complex numbers were presented and proved. Extensions to finite discrete fields would be interesting to the reviewer.

-Special suggestions: The introduction is correct. Comfortable to read. -The results are correct.

-Reviewer

Author Response

  See attachement

Reviewer 2 Report

Dear Author,

Thank you for your submitted manuscript.

In this paper, generalized complex numbers with respect to certain  functionals  are introduced. Also, an Euler type formula is derived. For random generalized complex numbers, moments are introduced and uniform distributions are analyzed.

I recommend the current paper for publication in Axioms. However, there are some corrections to be made:

- the order of references should be increasing;

-  "We not that" (line 148)

I think that some parts of the paper could be improved (for example the Section 2), and the references section extended. (I might add further suggestions later.)

Best wishes!

Author Response

See attachement

Reviewer 3 Report

Comments on “On Random Generalized Complex Numbers

The paper is well-written. However, the following must be incorporated in the revised form:

·       The introduction section should be enhanced to emphasize the study's motivation.

·       What is the research's major point of focus?

·       Do you think the topic is unique or relevant in the field?

·       What does it contribute to the topic matter relative to other published materials?

·       What specific methodological changes might the authors consider?

·       Are the citations appropriate? Check the reference section. References must be uniform.

·       Include any further remarks regarding the results and examples.

·       In the conclusion section, the authors should expand on or talk about all of the findings and the direction the paper will take in the future.

Author Response

See attachement

Round 2

Reviewer 3 Report

Authors addressed all the questions which was asked by reviewers' comments.

It can be accepted in the current form.

Author Response

Thank you.